# Radiomics for the Prediction of Response to Antifibrotic Treatment in Patients with Idiopathic Pulmonary Fibrosis: A Pilot Study

**DOI:** 10.3390/diagnostics12041002

**Published:** 2022-04-15

**Authors:** Cheng-Chun Yang, Chin-Yu Chen, Yu-Ting Kuo, Ching-Chung Ko, Wen-Jui Wu, Chia-Hao Liang, Chun-Ho Yun, Wei-Ming Huang

**Affiliations:** 1Department of Medical Imaging, Chi Mei Hospital, Tainan 710, Taiwan; vince7978@hotmail.com (C.-C.Y.); chency@seed.net.tw (C.-Y.C.); ytkuorad@gmail.com (Y.-T.K.); kocc0729@gmail.com (C.-C.K.); 2Department of Radiology, Faculty of Medicine, College of Medicine, Kaohsiung Medical University, Kaohsiung 807, Taiwan; 3Department of Health and Nutrition, Chia Nan University of Pharmacy and Science, Tainan 717, Taiwan; 4Institute of Biomedical Sciences, National Sun Yat-sen University, Kaohsiung 804, Taiwan; 5Division of Pulmonary and Critical Care Medicine, Mackay Memorial Hospital, Taipei 104, Taiwan; ybeei740317@gmail.com; 6Department of Biomedical Imaging and Radiological Sciences, National Yang Ming Chiao Tung University, Taipei 112, Taiwan; leehomliang@gmail.com; 7Department of Radiology, School of Medicine, College of Medicine, Taipei Medical University, Taipei 110, Taiwan; 8Department of Radiology, Wan Fang Hospital, Taipei Medical University, Taipei 116, Taiwan; 9Department of Radiology, Mackay Memorial Hospital, Taipei 104, Taiwan; 10Department of Medicine, Mackay Medical College, New Taipei City 252, Taiwan; 11Mackay Junior College of Medicine, Nursing, and Management, New Taipei City 252, Taiwan

**Keywords:** antifibrotic agents, disease progression, high-resolution computed tomography, idiopathic pulmonary fibrosis, interstitial lung disease, radiomics

## Abstract

Antifibrotic therapy has changed the treatment paradigm for idiopathic pulmonary fibrosis (IPF); however, a subset of patients still experienced rapid disease progression despite treatment. This study aimed to determine whether CT-based radiomic features can predict therapeutic response to antifibrotic agents. In this retrospective study, 35 patients with IPF on antifibrotic treatment enrolled from two centers were divided into training (*n* = 26) and external validation (*n* = 9) sets. Clinical and pulmonary function data were collected. The patients were categorized into stable disease (SD) and progressive disease (PD) groups based on functional or radiologic criteria. From pretreatment non-enhanced high-resolution CT (HRCT) images, twenty-six radiomic features were extracted through whole-lung texture analysis, and six parenchymal patterns were quantified using dedicated imaging platforms. The predictive factors for PD were determined via univariate and multivariate logistic regression analyses. In the training set (SD/PD: 12/14), univariate analysis identified eight radiomic features and ground-glass opacity percentage (GGO%) as potential predicators of PD. However, multivariate analysis found that the single independent predictor was the sum entropy (accuracy, 80.77%; AUC, 0.75). The combined sum entropy-GGO% model improved the predictive performance in the training set (accuracy, 88.46%; AUC, 0.77). The overall accuracy of the combined model in the validation set (SD/PD: 7/2) was 66.67%. Our preliminary results demonstrated that radiomic features based on pretreatment HRCT could predict the response of patients with IPF to antifibrotic treatment.

## 1. Introduction

Idiopathic pulmonary fibrosis (IPF) is a chronic, progressive, fibrosing lung disease of unknown etiology that primarily affects the elderly population [1]. It is characterized by an excessive fibrotic deposition in lung parenchyma by myofibroblasts that impedes gas exchange and ultimately leads to respiratory failure [1,2]. The overall prognosis is poor, and mortality usually occurs within 3–5 years after its diagnosis [3]. Fortunately, the availability of two antifibrotic agents, namely nintedanib and pirfenidone, has changed the landscape of care for IPF patients with improved clinical outcomes. Previous phase III trials (INPULSIS-1 and -2, ASCEND, and CAPCITY trials) and two recent network meta-analyses have consistently shown the remarkable effectiveness of both agents in slowing disease progression and reducing acute exacerbation (AE) compared to placebo [4,5,6,7,8]. Survival benefits have also been corroborated by real-world evidence from long-term cohort studies [9,10]. Nonetheless, the clinical course of IPF is variable and largely unpredictable, even after treatment [11]. A subset of patients treated with nintedanib (29.5%) or pirfenidone (12.6–21.0%) experienced an annual ≥10% decline in forced vital capacity (FVC), which is a strong predictor of mortality, upon completion of phase III trials [4,5,6,12]. Meanwhile, subgroup analyses found that the treatment effect is independent of baseline severity, sex, age, and smoking status [13,14]. As such, there is currently a knowledge gap regarding the determinants of therapeutic response.

Over the past decade, a number of prediction models for mortality has been developed and validated in IPF, such as the GAP (gender, age, and physiology variables) index and staging system [15]. Yet, no prediction model for therapeutic response has been proposed to date. To address this unmet need, high-resolution computed tomography (HRCT) may be a helpful tool as it plays a crucial role in diagnosing and monitoring IPF [1,16]. Furthermore, the extent of lung abnormalities on HRCT has been shown to correlate with lung function parameters and survival in these patients [17,18]. Preliminary studies revealed that certain CT findings, such as fibrosis and ground-glass opacities (GGOs), tend to increase more significantly in patients with unsatisfactory response [19,20]; in turn, these findings highlight the prognostic value of CT in determining their treatment response.

However, it can be challenging to evaluate and characterize the complex parenchymal abnormalities on HRCT with visual approach in the context of diffuse lung diseases including IPF. With the recent advances in computer-aided quantitative imaging techniques, radiologists can now minimize inter-and intraobserver variations in serial assessment, and improve their ability to capture the full extent of disease evolution [21,22]. Among these novel techniques, radiomics has emerged as a promising adjunct to clinical decision-making, which refers to the process of extracting quantitative imaging biomarkers from medical images through texture analysis using advanced computational algorithms [23,24]. These imaging biomarkers are also useful in predicting the clinical behavior and prognosis of various diseases [23,25,26,27,28], making them particularly important in the era of precision medicine. This study aimed to predict the treatment response of patients with IPF to antifibrotic agents using radiomics-based three-dimensional CT texture analysis.

## 2. Materials and Methods

### 2.1. Study Population

We retrospectively enrolled two separate cohorts of study patients from two tertiary care centers. The institutional review boards of both centers approved this study with waiver of informed consent. Cohort 1 consisted of 474 consecutive patients undergoing HRCT in Mackay memorial hospital, Taipei (institution 1), with suspected interstitial lung disease (ILD) from March 2010 to April 2021. Cohort 2 comprised 12 consecutive patients registered in the institutional pharmacy of Chi Mei hospital, Tainan (institution 2), for receiving antifibrotic agents from August 2020 to January 2022. The diagnosis of IPF in these patients was established through multidisciplinary discussion (MDD) as per the ATS/ERS/JRS/ALAT guidelines published in 2018 [16]. However, prior to the initiation of MDD team, diagnosis was made through consensus reached by pulmonologists and rheumatologists when definite or probable UIP pattern was evidenced on HRCT; for verification, each case was re-evaluated using the same criteria during data collection process. Only patients treated with nintedanib (150 mg twice daily) or pirfenidone (801 mg three times daily; maintenance dose) for at least three months were eligible for inclusion.

Exclusion criteria include: (1) no follow-up HRCT study or pulmonary function tests (PFTs) after treatment; (2) alternative diagnosis, or subtle fibrosis that was insufficient to establish the diagnosis of IPF; (3) presence of factors that could confound diagnosis and image analysis, including concomitant diseases (i.e., pneumonia, large pleural effusion, mass greater than 5 cm), or prior lung surgery.

### 2.2. Definition of Disease Progression

While IPF is, by definition, a progressive disease, a ≥10% decline in FVC over six to twelve months is generally considered significant and was used as a primary efficacy endpoint in drug trials because it portends an increased risk of mortality [12,29]. However, there is no consensus on defining disease progression for short-term evaluation. A recent meta-analysis identified an optimal cut-off of 5.7% over three-months to predict mortality with comparable performance to using the threshold of 10% at twelve-months [30]. With this background, we categorized patients into the progressive disease (PD) group if one of the following radiologic or functional criteria was met; if not, patients were categorized into the stable disease (SD) group. Functional criteria include: (1) FVC decline by ≥5.7% and ≥10% in patients who had been treated for 3–6 months and 6–12 months, respectively, upon outcome assessment; (2) AE reported by the pulmonologists; (3) all-cause mortality. Radiologic criteria include: (1) increased extent of fibrosis or decreased normally attenuated lung by ≥10%; (2) AE.

### 2.3. Acquisition Protocol for HRCT

In both institutions, all HRCT studies were carried out from the apex to the base of lung at end inspiration with the patients in supine position. Axial CT images (1.5-mm-thick) obtained within three months prior to the initiation of antifibrotic therapy were used for image analysis in this study. Detailed information of CT parameters is given in Appendix A.

### 2.4. Semiquantitative Fibrosis Quantification (Fibrotic Score)

Two experienced thoracic radiologists (C.-H.Y. and W.-M.H., with 19 and 9 years of expertise, respectively), who were blinded to clinical information, independently reviewed the non-enhanced CT images and rated the fibrotic scores. The fibrotic score is a 6-slices method proposed by Fraser et al. [31], which was based on cross-sectional CT images obtained at six anatomical levels: (1) aortic arch; (2) 1 cm below the carina; (3) right pulmonary venous confluence; (4) halfway between the third and fifth sections; (5) 1 cm above the right hemi-diaphragm; and (6) 2 cm below the right hemi-diaphragm (Figure 1). The total fibrotic score was calculated as the average percentage of three fibrotic components (honeycombing, reticulation, and GGO with traction bronchiectasis) of the six slices.

### 2.5. Whole-Lung CT Texture Analysis

Three-dimensional volumetric texture analysis of the HRCT images in DICOM (Digital Imaging and Communications in Medicine) format was implemented on a dedicated artificial intelligence (AI) platform (QUIBIM Precision 2.8, QUIBIM SL, Valencia, Spain). The automatic activation of quantitative analysis was based on the AI platform engine rule configured to verify if image DICOM tags matched predefined configurations for CT scans. The task of automated whole-lung segmentation was powered by 2-class U-Net-based Convolutional Neural Network (CNN) algorithm with deep supervision layers. This post-processing technique objectively quantifies imaging features related to lung parenchymal heterogeneity and pathology. Major categories of features were then extracted, including: (1) first-order statistics, which are derived from histogram of the voxel intensities; and (2) gray-level co-occurrence matrix (GLCM) statistics, which are second-order statistical measure of the spatial relationship of pixels that take into account the direction, local neighborhood and magnitude of changes [24]. The radiomics workflow is presented in Figure 2.

### 2.6. Automated Quantification of Parenchymal Patterns

Quantitative analysis of CT parenchymal patterns was carried out using an AI software (AVIEW, Coreline Soft, Seoul, Korea) established on content-based image retrieval (CBIR). Lung segmentation and disease pattern classification was performed with the use of a two-dimensional (2D) U-Net architecture with deep CNN through process previously described [32]. Automated parenchymal pattern analysis was classified following DICOM images successfully received by the AI platform. Each slice of HRCT was classified into six parenchymal patterns (honeycombing, reticulation, GGO, emphysema, consolidation, and normal lung) by validated CBIR system in pixel-level (Figure 3). To quantify the spatial distribution of parenchymal patterns, whole lung volume was divided into 64 cuboid (4 × 4 × 4) and regional fraction of each disease pattern in each cuboid was calculated after pattern extraction.

### 2.7. Statistical Analysis

All statistical analyses were performed using SPSS (v22.0, IBM, Chicago, IL, USA). Continuous and categorial variables of the clinical data and baseline PTFs between the SD and PD groups were compared using Mann-Whitney U test and Fisher’s Exact test, respectively. Intraclass correlation coefficient (ICC) was used to examine inter-rater variability of fibrotic score measurements. Univariate and multivariate logistic regression analyses were used to determine the predictors of PD. Variables with *p*-value < 0.1 in the univariate analysis were included in the multivariate analysis [33], whereas only covariates with a two-tailed *p*-value < 0.05 in the multivariate analysis were regarded as the independent predictor. The predictive performance was validated by receiver operating characteristic (ROC) curve.

## 3. Results

### 3.1. Study Population

Training set—Of the initially screened 474 patients in cohort 1, 358 patients were excluded owing to the following reasons: alternative diagnosis (*n* = 190), presence of subtle fibrosis (*n* = 103), prespecified concomitant diseases (*n* = 60), and prior lung surgery (*n* = 5). The diagnosis of IPF was confirmed in 116 patients; among them, only 26 patients underwent antifibrotic treatment for at least three months and were finally included in the training set (SD/PD: 12/14). 

Validation set—Among the 12 patients who underwent antifibrotic therapy in cohort 2, nine patients were included (SD/PD: 7/2) after the exclusion of three patients due to alternative diagnosis (*n* = 1; systemic sclerosis-related ILD), treatment duration shorter than three months (*n* = 1), and lung cancer greater than 5 cm (*n* = 1).

The selection process and clinical characteristics of the study patients are presented in Figure 2 and Table 1, respectively. There was no significant imbalance between the SD and PD groups in the two datasets based on baseline pulmonary function parameters, mean treatment duration, and smoking history. However, the GAP index and stage were significantly higher in the PD groups compared to the SD group in the training set.

### 3.2. Inter-Rater Reliability of the Fibrotic Score 

The ICC was 0.91, and an ICC > 0.9 was considered excellent. In the training set, there was no significant difference in the mean fibrotic scores of the SD and PD groups (*p* < 0.1; Table 1).

### 3.3. Radiomic Feature Extraction and Selection 

In total, 26 first-and second-order (GLCM) radiomic features were extracted (Table 2). In univariate logistic regression analysis, eight features showed significant differences (*p* < 0.1) between the SD and PD groups, which include entropy, difference entropy, sum entropy, kurtosis, skewness, inverse difference, and maximum probability. However, only the sum entropy remained statistically significant (OR = 0.01, *p* < 0.05) in multivariate logistic regression analysis (Table 3 and Figure 4).

### 3.4. Automated Quantification of Parenchymal Patterns 

Of the six quantified radiologic patterns, only the GGO percentage (GGO%) was significantly different (*p* < 0.1) between the SD and PD groups based on univariate logistic regression analysis; however, this was not statistically significant (*p* < 0.05) in the multivariate logistic regression analysis (Table 3).

### 3.5. Development of Prediction Model with Performance Evaluation and Validation

In the training set, the AUCs of the models built with radiomic feature (sum entropy) and quantified parenchymal pattern (GGO%) were 0.75 (95% CI, 0.54–0.89; sensitivity, 100.00%; specificity, 58.33%; accuracy, 80.77) and 0.69 (95% CI, 0.48–0.85; sensitivity, 57.14%; specificity, 75.00%; accuracy, 65.38%), respectively. The combined sum entropy-GGO% model yielded an AUC of 0.77 (95% CI, 0.56–0.91; sensitivity, 100.00%; specificity, 75.00%; accuracy, 88.46%), which was higher than the values found when using any of the models alone (Table 4 and Figure 5). In contrast, the GAP model showed comparable values of AUC (0.73–0.77) but a lower accuracy (73.07%). In the validation set, although the accuracy of the combined model (66.67%) did not differ from that of the radiomic feature (sum entropy), it was higher than that of the GAP model (37.50%) (Table 5).

## 4. Discussion

Radiomics yields a plethora of objective quantitative features that entail prognostic implications. It is also widely recognized as a valuable tool to guide individualized patient care. In this study, univariate logistic regression analysis showed that the selected features—sum entropy, entropy, difference entropy, kurtosis, skewness, inverse difference, and maximum probability—enabled the prediction of PD after treatment, whereas multivariate logistic regression analysis showed that the single independent predictor of PD was the sum entropy, which demonstrated an accuracy of 80.77% (AUC, 0.75). On the other hand, while the quantified parenchymal pattern (GGO%) using AI software also predicts the PD based on univariate analysis, this was not observed in multivariate analysis. Regardless, this variable was not eliminated because our results were similar to those of Balestro et al., who suggested that GGO may be associated with unsatisfactory treatment response based on a visual semiquantitative analysis of temporal changes in CT after six months [19]. Interestingly, the combined sum entropy-GGO% model improved the overall predictive performance (accuracy, 88.46%; AUC, 0.77). Although originally developed for predicting mortality in patients with IPF, the GAP model was compared with our proposed models for the prediction of PD. The results showed that both the sum entropy and combined model outperformed the GAP model by achieving higher accuracy (80.77–88.46% vs. 73.07%), albeit with comparable AUCs. Similarly, both the sum entropy and combined model also showed a higher accuracy of 66.67% as compared to the GAP model (37.50%) on external validation. To the best of our knowledge, this study across two cohorts of IPF patients first demonstrated that CT-based radiomics can be used to predict their response to antifibrotic agents. Furthermore, automated whole-lung segmentation obviates subjective variations in manual ROI selection; hence, it offers better reproducibility.

Previous studies showed that radiomics enhances the diagnosis and severity staging of IPF [25,26], and predicts the functional decline and long-term survival of patients [27,28]. Notably, our previous work demonstrated that a CT-based radiomics model (specifically, energy and kurtosis) is capable of predicting lung cancer development in patients with IPF [34]. In the present study, certain radiomic features that distinguish between the PD and SD groups are in line with prior studies, including entropy-related features and kurtosis. Entropy is a measure of the inherent randomness in gray level intensities whereas sum entropy represents the sum of intensity differences of the neighborhood [24]; these two statistics have recently garnered much attention in oncology because they reflect tumor heterogeneity and are predictive of tumor stage and prognosis [35]. IPF is considered a cancer-like disease because it is marked by an uncontrolled fibroproliferation akin to neoplasia, and both diseases share several common genetic alterations and molecular pathways in pathogenesis [36]. We speculate that the higher entropy-related features observed in the PD group might reflect greater distortion of lung parenchyma caused by myofibroblasts in a manner similar to tumor cells. Kurtosis refers to a first-order statistic that measures the peakedness of intensity distribution. A low kurtosis, which represents an intensity distribution toward the two ends of high (fibrosis) and low (destructive lucent center of honeycombing) in the context of IPF [22], was shown to predict the disease progression and short-term mortality [27]. 

The past decade of research has focused largely on the impact of antifibrotic therapy on lung function; thus, little is known about the consequent changes in imaging or histopathology during follow-up after treatment. In a phase IIIb placebo-controlled trial, Lancaster et al., demonstrated that the increment in reticulation was smaller in the nintedanib group compared to that in the placebo group after six months using another quantitative texture analysis tool (computer-aided lung informatics for pathology evaluation and rating, CALIPER) [37]. Zhang et al., studied 28 patients with IPF undergoing lung transplantation and suggested that treated patients tend to have less histologic evidence of acute lung injury [38]. In a recent histologic analysis of 40 nintedanib-treated patients, Nemoto et al., found that edematous changes in the interlobular septum may be associated with poor response [2]. On the other hand, Jacob et al., reported that among CALIPER-derived parameters, the pulmonary vessel-related structure (VRS) score obtained from the upper-and mid-lung zones was the strongest predictor of 10% FVC decline or death at 12-months after the treatment [39]. 

An early prediction of therapeutic efficacy is beneficial in tailoring treatments for patients with IPF. Since the most common adverse effects of nintedanib (diarrhea, 62%) and pirfenidone (nausea, 36.8%) tend to occur in the first three months [40], this could result in an early withdrawal from therapy (4.4% of the cases in INPULSIS trials) [4]. Worst of all, treatment discontinuation is associated with a greater risk of death and potentially leads to an accelerated disease progression [40,41]. Our prediction model may also help strengthen the medication adherence particularly in patients who are likely to have favorable outcomes, and optimize the risk stratification of vulnerable subjects with unsatisfactory response who may require additional therapy.

This study has some limitations. First, given the rarity of IPF in Asian populations and the common, yet controversial, practice of withholding antifibrotic treatment from functionally stable subjects [1,9], the patient number is relatively small even though the training set includes a cohort of patients with IPF over a decade. The small sample size may render the prediction model prone to bias. In light of this, external validation was implemented to assess the reproducibility and generalizability of the model, and it showed an acceptable accuracy in predicting the PD. Second, selection bias could occur because the diagnosis of IPF was not established through MDD for all patients. Third, although a previous study suggested that VRS predicts PD [39], this study was unable to compare VRS with GGO since the software we used could not quantify this parameter. Finally, the length of antifibrotic therapy until outcome assessment was not uniform among the patients (ranging from 3–12 months); nonetheless, the definition of disease progression was modified based on the current evidence [12,29,30], and a significant imbalance between the SD and PD groups was not found in both the training and validation sets.

## 5. Conclusions

In this pilot study, a pretreatment HRCT-based radiomics model was developed to predict the response to antifibrotic treatment in patients with IPF. The preliminary results demonstrated that the sum entropy showed a high accuracy in predicting the PD and that the combined sum entropy-GGO% model further improved the overall predictive performance. These findings provide potential clinical benefits as they may allow the clinicians to tailor treatment strategy to individual patients. Future studies on larger-scale datasets are needed to corroborate our conclusions and to examine the robustness of the model.

## Figures and Tables

**Figure 1 diagnostics-12-01002-f001:**
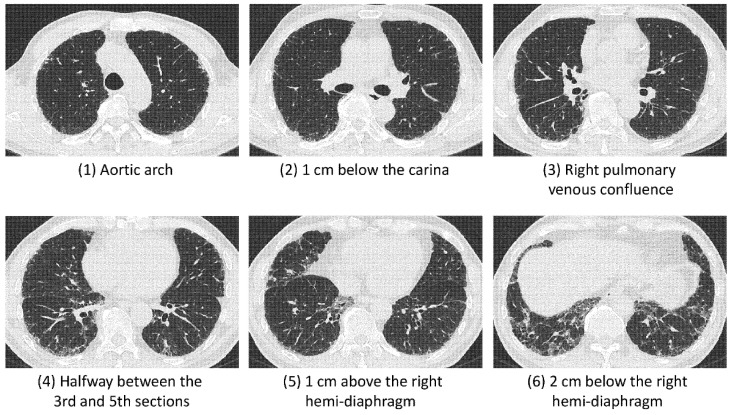
Axial thin-section HRCT images of a 66-year-old male patient diagnosed with IPF through MDD with a (definite) UIP pattern. The total fibrotic score was calculated as the average percentage of the three fibrotic components (honeycombing, reticulation, and GGO with traction bronchiectasis) of the six slices.

**Figure 2 diagnostics-12-01002-f002:**
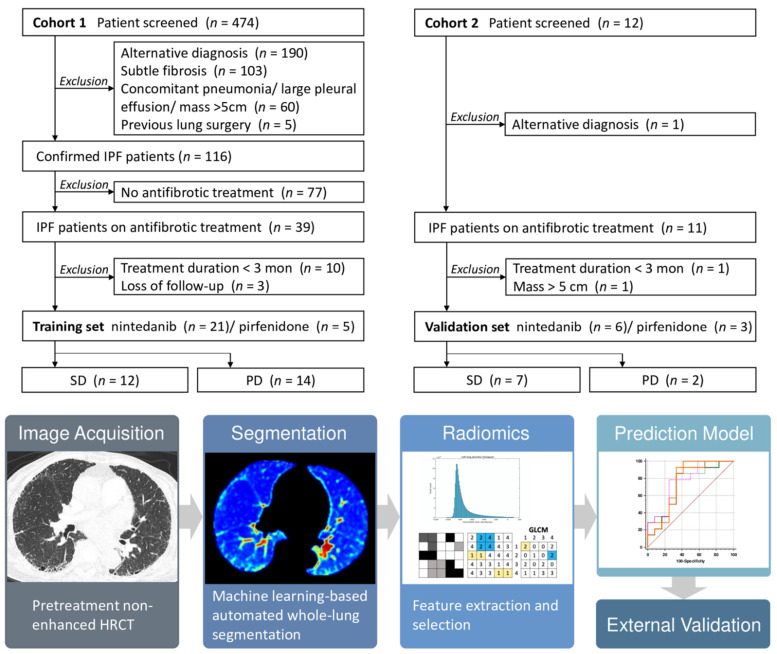
Flow diagrams of patient selection process and radiomics workflow.

**Figure 3 diagnostics-12-01002-f003:**
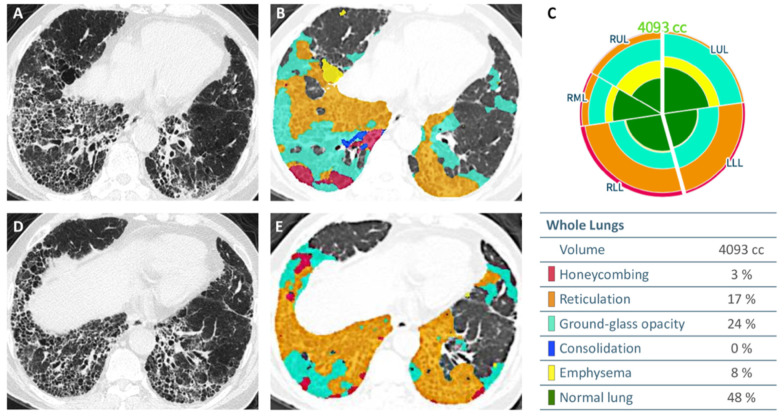
Automated quantification of CT parenchymal patterns in a 74-year-old male patient with IPF. Axial thin-section HRCT images obtained at the lung base, without (**A**,**D**) and with (**B**,**E**) overlay of color maps depicting the distribution and extent of different parenchymal abnormalities. The glyph and chart (**C**) summarize the percentage of the six parenchymal patterns.

**Figure 4 diagnostics-12-01002-f004:**
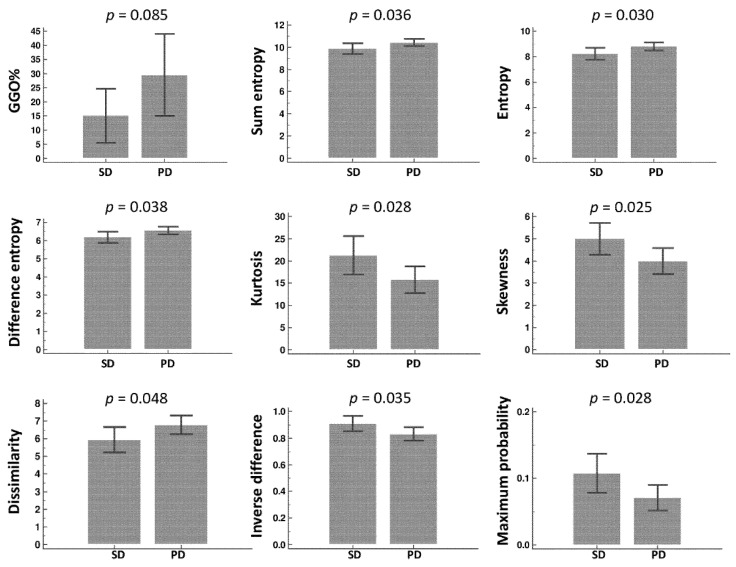
Comparison of the selected radiomic features and GGO% in the SD and PD groups.

**Figure 5 diagnostics-12-01002-f005:**
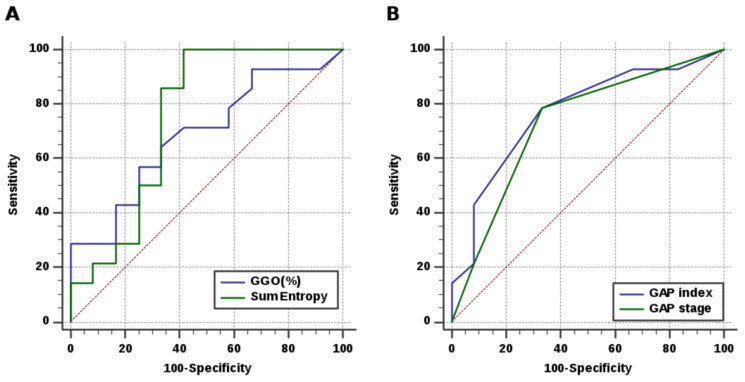
ROC curves of the prediction models for progressive disease. (**A**) The radiomic feature (sum entropy) demonstrated an acceptable AUC of 0.75 which was higher than that of the quantified parenchymal pattern (GGO%; AUC, 0.69), but was not superior to that of (**B**) the GAP index and stage (AUC, 0.77 and 0.73, respectively). The combined sum entropy-GGO% model improved the predictive performance with an AUC of 0.77.

**Table 1 diagnostics-12-01002-t001:** Comparison of Patient Characteristics in the Training and Validation Sets.

Characteristics	Training Set			Validation Set		
	SD (*n* = 12)	PD (*n* = 14)	*p*-Value	SD (*n* = 7)	PD (*n* = 2)	*p*-Value
Age	72.58 ± 10.21	76.14 ± 7.88	0.326	76.71 ± 7.99	69.00 ± 0.00	0.018 *
Sex (M)	7 (58%)	12 (86%)	0.139	6 (86%)	0	0.043 *
Smoking	5 (42%)	7 (50%)	0.686	4 (57%)	0	0.030 *
PFTs						
FVC (%)	82.17 ± 19.69	82.09 ± 22.80	0.993	91.57 ± 26.61	65.00	0.386
FEV1 (%)	90.42 ± 20.37	79.90 ± 30.58	0.321	94.43 ± 20.35	73.00	0.363
DLCO (%)	69.36 ± 19.71	62.45 ± 17.27	0.392	63.33 ± 33.63	54.00	0.807
TLC (%)	75.50 ± 8.94	72.07 ± 12.03	0.425	89.29 ± 20.23	60.00	0.224
GAP index	3.00 ± 1.41	4.43 ± 1.60	0.025 *	4.00 ± 1.26	4.50 ± 0.71	0.625
GAP stage	1.42 ± 0.67	2.00 ± 0.68	0.038 *	2.00 ± 0.63	2.00 ± 0.00	1.000
Treatment duration (weeks)	48.95 ± 17.60	35.96 ±18.64	0.082	36.96 ± 14.48	16.50 ± 2.12	0.099
Fibrotic score	19.89 ± 10.59	25.50 ± 11.08	0.200	21.90 ± 9.06	26.72 ± 23.65	0.639
Lung volume (mL)	3242.25 ± 666.33	3057.29 ± 849.69	0.546	3308.96 ± 1362.99	3553.67 ± 49.14	0.816

Values are given as mean ± standard deviation. * Indicates statistical significance. Abbreviation: DLCO, diffusing capacity of the lung for carbon monoxide; FEV1, forced expiratory volume in 1 s; FVC, forced vital capacity; GAP, gender-age-physiology index and staging system; PD, progressive disease group; PFTs, pulmonary function tests; SD, stable disease group; TLC, total lung capacity.

**Table 2 diagnostics-12-01002-t002:** Comparison of Radiomic Features of Stable Disease (SD) and Progressive Disease (PD) Groups in the Training Set.

Metrics	Features	SD	PD	*p*-Value
First order	Energy	2.21 × 10^12^	2.03 × 10^12^	0.385
	Entropy	8.23	8.80	0.030 *
	Kurtosis	21.28	15.77	0.028 *
	Skewness	5.00	4.00	0.025 *
	Mean	−411.04	−357.93	0.415
	Standard deviation	359.54	382.53	0.169
	Median	−530.11	−479.29	0.409
	10th percentile	−678.26	−675.09	0.951
	90th percentile	13.85	137.23	0.213
Second order (GLCM)	Autocorrelation	444.15	556.28	0.075
	Cluster prominence	627,504.74	590,318.29	0.651
	Cluster shade	10,774.07	10,488.91	0.820
	Contrast	50.01	58.24	0.101
	Correlation	1.45	1.44	0.788
	Difference entropy	6.19	6.55	0.038 *
	Difference variance	30.55	33.30	0.247
	Dissimilarity	5.95	6.79	0.048 *
	Inverse difference	0.91	0.83	0.035 *
	IMC1	−0.28	−0.28	0.837
	IMC2	1.55	1.56	0.642
	Maximum probability	0.11	0.07	0.028 *
	Sum average	54.13	61.10	0.059
	Sum entropy	9.87	10.45	0.036 *
	Sum of squares	93.72	107.59	0.160
	Sum variance	324.86	372.13	0.190

* Indicates statistical significance. Abbreviation: GLCM, grey level co-occurrence matrix; IMC, information measure of correlation.

**Table 3 diagnostics-12-01002-t003:** Univariate and Multivariate Logistic Regression Analyses to Differentiate the Progressive Disease (PD) Group from the Stable Disease (SD) Group in the Training Set.

Characteristics	Univariate Regression Analysis			Multivariate Regression Analysis		
	OR	95% CI	*p*-Value	OR	95% CI	*p*-Value
Entropy	4.37	1.05–18.30	0.04 *	3.42 × 10^75^	0.02–5.94 × 10^153^	0.06
Difference entropy	8.15	0.99–66.94	0.05 *	1.67 × 10^16^	0.01–4.14 × 10^40^	0.19
Sum entropy	3.93	1.01–15.32	0.05 *	0.01	0.01–0.22	0.04 *
Kurtosis	0.85	0.73–1.01	0.04 *	0.90	0.25–3.25	0.87
Skewness	0.40	0.16–0.95	0.04 *	0.01	0.01–63.14	0.29
Dissimilarity	2.30	0.97–5.48	0.06 *	0.01	0.01–525.21	0.16
Inverse difference	0.03	0.01–0.95	0.05 *	1.40 × 10^61^	0.35–5.58 × 10^122^	0.05
Maximum probability	0.02	0.01–0.47	0.04 *	0.01	0.01–2.21 × 10^42^	0.58
GGO%	1.04	0.97–1.09	0.09 *	1.10	0.99–1.22	0.07
Honeycombing%	0.75	0.21–2.73	0.67			
Reticulation%	1.06	0.84–1.34	0.62			
Emphysema%	1.04	0.89–1.13	0.92			
Age	1.08	0.98–1.19	0.13			
Sex	4.29	0.65–28.26	0.13			
Smoking	1.40	0.30–6.62	0.67			

* Indicates statistical significance.

**Table 4 diagnostics-12-01002-t004:** Prediction Performance of Different Models in the Training Set.

Characteristics	Cut-Off	AUC	Sensitivity (%)	Specificity (%)	Accuracy (%)
Entropy	>7.95	0.76 [0.55–0.90]	100.0	58.33	80.77
Difference entropy	>6.39	0.74 [0.54–0.89]	78.57	75.00	76.92
Sum entropy	>9.60	0.75 [0.54–0.89]	100.00	58.33	80.77
Kurtosis	≤19.45	0.73 [0.52–0.88]	85.71	66.67	76.92
Skewness	≤4.98	0.76 [0.56–0.91]	92.86	66.67	80.77
Dissimilarity	>5.62	0.76 [0.55–0.90]	92.86	58.33	76.92
Inverse difference	≤0.90	0.77 [0.56–0.91]	78.57	75.00	76.92
Maximum probability	≤0.09	0.74 [0.54–0.89]	85.71	66.67	76.92
GGO%	>16.00	0.69 [0.48–0.85]	57.14	75.00	65.38
Sum entropy + GGO%		0.77 [0.56–0.91]	100.00	75.00	88.46
GAP index	>3	0.77 [0.56–0.91]	78.57	66.67	73.07
GAP stage	>1	0.73 [0.52–0.88]	78.57	66.67	73.07

**Table 5 diagnostics-12-01002-t005:** Prediction Accuracy of Different Models in the Validation Set Using the Cut-Off Values Derived from the Training Set.

Characteristics	Sensitivity (%)	Specificity (%)	Accuracy (%)
Entropy	50.00	71.43	66.67
Difference entropy	50.00	85.71	77.78
Sum entropy	50.00	71.43	66.67
Kurtosis	50.00	42.86	44.44
Skewness	50.00	28.57	33.33
Dissimilarity	50.00	85.71	77.78
Inverse difference	50.00	71.43	66.67
Maximum probability	50.00	71.43	66.67
GGO%	50.00	50.00	50.00
Sum entropy + GGO%	50.00	74.43	66.67
GAP index	100.00	16.67	37.50
GAP stage	100.00	16.67	37.50

## Data Availability

Not applicable.

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
