# Peer review of "Radiomics for the Prediction of Response to Antifibrotic Treatment in Patients with Idiopathic Pulmonary Fibrosis: A Pilot Study"

_diagnostics, 2022, doi:10.3390/diagnostics12041002_

Round 1

Reviewer 1 Report

The authors presented here a work studying the interest of radiomics combined with artificial intelligence, as a marker of response to antifibrosis therapies in patients with idiopathic pulmonary fibrosis.

The interest of the study is strong in view of the severity of the pathology and the heterogeneity of the therapeutic response of patients.

It is a very well conducted study of high syntactic and scientific quality.

Here are some minor remarks to further improve the manuscript which is, as it is, of very good quality.

1) How many lung biopsies were necessary to assess the diagnosis?
2) Could you implement the GAP index in your algorithm, or at least discuss it?
3) Could you precise wich antifibrotic therapy (nintedamib? pirfenidone?) were administred ?
4) It would be interesting to assess, as a supplementary data,  if patients with a pejorative IA score more often underwent a severe acutisation in their medical history, or not.

Reviewer 2 Report

The authors have reported a pilot study on radiomics for the prediction of response to antifibrotic treatment in patients with Idiopathic Pulmonary Fibrosis which  demonstrated that radiomic features based on pretreatment HRCT could predict the response of patients with IPF to antifibrotic treatment. The study is interesting. In general, the main conclusions presented in the paper are supported by the figures and supporting text. However, to meet the journal quality standards, the following comments need to be addressed.

  1. The introduction can be improved. The authors should focus on extending the novelty of the current study. There are two many information of data collection details which should be shortened. Emphasize should be given in improvement of the  model compared to   existing  state-of-the art models.

  1. The introduction writing part need to be improved. Also, the writing and presentation of the introduction lacks a bit in clarity. The paper requires a some amount of rewriting to clarify all aspects of it, especially the novelty and new findings of this work that need to be clearly mentioned. The authors mentioned—“There is currently a knowledge gap regarding the determinants of therapeutic response. This could potentially be addressed by high-resolution computed tomography 61 (HRCT), as it plays a crucial role in diagnosing and monitoring IPF’ this point needs to be elaborated with existing research work aligned to this direction.

  1. Page 4: “The task of automated whole-lung segmentation was powered by 2-class U-Net- 145 based Convolutional Neural Network (CNN) algorithm with deep supervision layers.” Please provide the details of the model networks and why it has  been chosen and what modification has been performed compared to state-of-the-art models.

  1. N=39 seems very low numbers of patience. Did the authors employ any data augumentation/ enhancement methods before training? If so, it should be mentioned.

  1. Fig 3: only one segmentation result has been provided. Please add more results to see the robustness of the model.
  2. Table 1 . What are the baseline models and benchmark results? The authors can compare the result from various models evaluated in same  datasets?
  3. Also, all hyperparameters (learning rate, mini-batch size, number of epochs, optimizer) and model complexity should be detailed
  4. what about comparison of the result with current state-of-the art models? Did authors perform ablation study to compare with different models?
  5. The authors should provide state-of-the-art deep learning algorithms relevant to the study  (see Sensors 2021, 21(9), 3263; https://doi.org/10.3390/s21093263, Comp  Elect in Agri (2022), 193 106694 https://doi.org/10.1016/j.compag.2022.106694,). Hence it should be mentioned.
  6. Conclusion parts needs to be strengthened.
  7. Please provide a fair weakness and limitation of the model, and how it can be improved.
  8. Typographical errors: There are several minor grammatical errors and incorrect sentence structures. Please run this through a spell checker.

Reviewer 3 Report

I would like to congratulate the authors for their interesting and informative paper. This is a retrospective study investigating the role of CT-based radiomic features in the prediction of therapeutic response to antifibrotic agents for idiopathic pulmonary fibrosis. The study included 35 patients from two institutions. The authors found that certain radiomic features based on pre-treatment high resolution CT can potentially predict the response of patients with idiopathic pulmonary fibrosis to antifibrotic treatment.

Here, I have made a few suggestions that, in my opinion, could help improve the overall quality of the manuscript.

  • Unfortunately, the number of included patients is relatively small, rendering the study prone to relevant bias. Although the authors recognise that as a limitation of their study, it may worth further emphasizing and discussing it.
  • The authors may consider further clarifying the details of the antifibrotic treatments administered to the patients.
  • The authors may consider discussing the significant differences in some of the baseline characteristics of the patients in the validation set, and how these could affect the results, if at all.

Round 2

Reviewer 2 Report

Although, the authors did address most of the reviewer's previous comments, however, some of the comments, in particular, comments: 3, 9 were not satisfactorily addressed. In my opinion, the new version of the manuscript needs further revision before publication.

Replay to response  3: The reviewers were asked to elaborate on the novelty of the model  and general applicability of the study in comment 1. This is very important. However, seems like the authors have used "Quibim’s automated Whole Lung Parenchyma Texture Analysis software". Then the novelty of this study is quentionable. The authors should at least aware of hyperparameters of the model (as asked in comment 7). That should be provided.  Particularly, the authors should provide current state-of-the-art works as mentioned previously by the reviewer. So that the general reader should be aware of it.  Hence should be addressed in the introduction.

Replay to response  9 : Additionally, I did not see any discussion regarding state-of-the-art deep learning algorithms as mentioned which can be extended for predicting the treatment response of patients with IPF to antifibrotic agents. It needs to be addressed.

 The authors should take care of the points raised by the reviewer to improve the quality of the papers and for further consideration. The authors should clearly highlight the necessary changes in the main manuscript.
